# WiPP: Workflow for Improved Peak Picking for Gas Chromatography-Mass Spectrometry (GC-MS) Data

**DOI:** 10.3390/metabo9090171

**Published:** 2019-08-21

**Authors:** Nico Borgsmüller, Yoann Gloaguen, Tobias Opialla, Eric Blanc, Emilie Sicard, Anne-Lise Royer, Bruno Le Bizec, Stéphanie Durand, Carole Migné, Mélanie Pétéra, Estelle Pujos-Guillot, Franck Giacomoni, Yann Guitton, Dieter Beule, Jennifer Kirwan

**Affiliations:** 1Core Unit Bioinformatics, Berlin Institute of Health, 10178 Berlin, Germany; 2Berlin Institute of Health Metabolomics Platform, 10178 Berlin, Germany; 3Max Delbrück Center for Molecular Medicine in the Helmholtz Association, 13125 Berlin, Germany; 4Integrative Proteomics and Metabolomics, Berlin Institute for Medical Systems Biology, Max Delbrück Center for Molecular Medicine, 13125 Berlin, Germany; 5Charité—Universitätsmedizin Berlin, 10178 Berlin, Germany; 6Université Clermont Auvergne, INRA, UNH, Plateforme d’Exploration du Métabolisme, MetaboHUB Clermont, 63000 Clermont-Ferrand, France; 7LABERCA, Oniris, INRA, Université Bretagne—Loire, 44307 Nantes, France

**Keywords:** gas chromatography-mass spectrometry (GC-MS), metabolomics, peak detection, peak classification, pre-processing workflow, parameter optimisation, machine learning, support vector machine

## Abstract

Lack of reliable peak detection impedes automated analysis of large-scale gas chromatography-mass spectrometry (GC-MS) metabolomics datasets. Performance and outcome of individual peak-picking algorithms can differ widely depending on both algorithmic approach and parameters, as well as data acquisition method. Therefore, comparing and contrasting between algorithms is difficult. Here we present a workflow for improved peak picking (WiPP), a parameter optimising, multi-algorithm peak detection for GC-MS metabolomics. WiPP evaluates the quality of detected peaks using a machine learning-based classification scheme based on seven peak classes. The quality information returned by the classifier for each individual peak is merged with results from different peak detection algorithms to create one final high-quality peak set for immediate down-stream analysis. Medium- and low-quality peaks are kept for further inspection. By applying WiPP to standard compound mixes and a complex biological dataset, we demonstrate that peak detection is improved through the novel way to assign peak quality, an automated parameter optimisation, and results in integration across different embedded peak picking algorithms. Furthermore, our approach can provide an impartial performance comparison of different peak picking algorithms. WiPP is freely available on GitHub (https://github.com/bihealth/WiPP) under MIT licence.

## 1. Introduction

Metabolomics and related sciences use a combination of analytical and statistical approaches to qualitatively and quantitatively analyse the small molecules in a cell or biological system to answer biological questions [1,2]. Metabolomics benefits from maximizing the number of compounds detected in any individual analysis, while requiring concurrently that the results are robust and reproducible. Gas chromatography-mass spectrometry (GC-MS) is a common technology used in metabolomics research and contains information on both the chromatographic and mass spectral space of the data [3]. Before downstream statistical and functional analysis, the data must first be pre-processed, such that individual peaks are identified, retained, and catalogued in a numerical format, while irrelevant noise data should be removed. Attempting this by hand is a laborious process unsuited for epidemiological size datasets and also impedes reproducibility of this important data analysis step. Instead, this is commonly achieved by using one of many software options on the market, e.g., XCMS [4], metaMS [5], MetAlign [6], mzMine [7], ADAP-GC [8,9], PyMS [10], and eRah [11]. It is commonly understood that there are still certain conditions in which these automated methods are sub-optimal, and the user-defined settings, as well as some of the hard coded features of each software will have a large impact on the results [12]. However, each tends to have its strengths and weaknesses and will result in a slightly different result [8,12]. In this study, we sought to combine the strengths of different algorithms while minimizing the weaknesses.

In order to benefit from the strengths of each individual peak picking algorithm, we adopted a machine learning approach to classify peaks, enabling the automatic optimisation of user-definable parameters for each algorithm and the combining of results. Machine learning uses statistical and pattern recognition strategies to progressively improve their learning of data interpretation without requiring specific data interpretation programming [13]. Various forms of machine learning have previously been used for metabolomics studies, including a least squares-support vector machine, a support vector machine regression, and random forest and artificial neural networks [14,15,16]. Support vector machines (SVM) is a well-known supervised machine learning models, which are well suited for classification analysis [17]. Supervised learning uses an existing classified dataset(s) to train the model. This has the advantage that the resulting model is easy to optimise and validate [13].

We present a novel approach to automate peak picking in GC-MS data in order to optimise the reproducibility, accuracy, and quality of the process by combining the strengths of multiple existing or new peak picking algorithms. We apply a visualization strategy combined with a support vector machine (SVM)-based supervised machine learning approach to assess and learn peak quality. The learned model enables us to perform automatic optimisation of parameters for different peak picking algorithms. This is achieved by scoring the algorithm results on a representative data set. The learned model is also used to evaluate and integrate peaks suggested by two or more embedded (pre-existing) peak picking algorithms. The workflow result is a single high quality, high confidence peak set, which is suitable for immediate further downstream analysis. Once workflow for improved peak picking (WiPP) has been trained for a particular setting (resolution and sample type), it can quickly process large data sets automatically without further user involvement. The WiPP output format is designed to be easily searched by mass spectral library software, such as NIST, and can be subsequently analysed using standard statistical tools. It consists of a .csv and a .msp file, detailing the individual peaks as chromatographic retention time or retention index-mass spectral fragment groups and their associated individual absolute mass spectral peak intensities.

## 2. Results

To demonstrate and evaluate the performance of the WiPP method, we determined the recovery of compounds from a known mix at different concentrations (Section 2.1). To quantify the added value, we also compared WiPP results with those obtained from the individual algorithms used within WiPP. Furthermore, we performed a case study using a complex biological data set: We compared results obtained using WiPP to a published independent analysis by a third party (Section 2.2). Readers less familiar with peak detection, algorithm benchmarking, and machine learning approaches might benefit from first reading Section 4.1 of Material and Methods, which explains important concepts and defines relevant terms and necessary technical jargon.

### 2.1. Validation and Benchmarking

A known mix of commercially available standards was analysed at three different concentrations. Datasets were acquired on two different GC-MS instruments using different (high and low) resolutions using multiple replicates each (Section 4.3.1). High quality peak sets were generated using the WiPP implementation, utilising centWave and matchedFilter peak picking algorithms. The full pipeline was run, including a manual classification of training sets to generate SVM peak classifiers, parameter optimisation for both algorithms, and the filtering step, as described by the methods section. The sets of optimal parameters determined for each peak picking algorithm and both datasets are available in Appendix A. For comparison, the parameter optimisation of matchedFilter was also performed on the high resolution data using IPO [18], a tool optimising parameters based on isotopologue detection. Overall, similar optimal parameters were found with notable exceptions as shown in Appendix A. Possible reasons for these exceptions are explored in the discussion section.

WiPP removes peaks classified with insufficient and intermediate quality from the final high-quality peak set but always keeps them accessible to the user in a separate file. The following analysis was conducted only on the high-quality peak set, i.e., it does not consider the peaks of intermediate quality classes, because we do not consider their quantification fit for immediate downstream processing.

First, we compared the performance of WiPP with the embedded matchedFilter and centWave algorithm alone, always using the algorithm parameters determined by WiPP. The total number of detected peaks and their classification into either high quality or intermediate or low quality were analysed and contrasted (Figure 1). For example, in the high concentration, low resolution dataset 1, the total number of peaks detected was 137 for matchedFilter and 238 for centWave, of which 88 (59.9%) and 144 (60.5%) unique peaks were respectively classified as high quality. By contrast, in the high concentration, high resolution dataset 2, the total number of peaks detected was 2153 for matchedFilter and 997 for centWave, of which 280 (13.0%) and 181 (18.2%) unique peaks were respectively classified as high quality. Figure 1A shows that, in low resolution data, the number of peaks annotated as high quality and detected by centWave is on average 47% higher than the number of high-quality peaks detected by matchedFilter. However, the output of the two algorithms do not entirely overlap, and there are cases where matchedFilter detects high quality peaks which centWave does not and vice versa. Therefore, the number of peaks classified as high-quality increases by merging the results of both algorithms. As the compound mix concentration increases, the number of high-quality peaks found by both individual algorithms and in WiPP increases due to a reduced number of peaks near the noise level. The number of peaks filtered out by WiPP (Figure 1B) in low resolution data for matchedFilter and centWave represents on average of 40% and 43% of the total number of peaks reported by the two algorithms, respectively.

Similarly, as for low resolution data, the use of centWave and matchedFilter together increases the number of high-quality peaks detected in high resolution datasets. However, matchedFilter detects 44% more high quality peaks in comparison to centWave (Figure 1C). The number of filtered peaks is very high in comparison to low resolution data, as they represent an average of 90% of the total number of peaks detected by matchedFilter and 80% of the total number of peaks detected by centWave (Figure 1D).

Appendix A explores the performance of the SVM classifier with respect to the size of the training data set. Please note that standard ROC curves are not applicable for the classifier evaluation due to the multi-class classification approach, i.e., the classification does not depend on a single threshold that could be varied to generate different sensitivity and recall rates. Therefore, we generated confusion matrices (Appendix A) to assess if certain peak classes were misclassified systematically. The highest number of constantly misclassified peaks was found for matchedFilter in high resolution data, where 9% of peaks were wrongly classified as merged/shouldering and 10% of noise peaks were classified differently. The misclassified peaks were evenly distributed among the other classes, which lowers the overall result but does not distort it systematically.

Compound detection is not a built-in feature of WiPP, but the WiPP output format enables easy library matching using existing compound libraries. To test the ability of WiPP to report true compound related peaks, peaks detected and classified as high quality were annotated using our internal library, corresponding to the compound mix using reverse matching (Appendix A). The output of the automated annotation implemented in WiPP was separated into two categories: High confidence annotation, requiring both the retention index (RI) to be within a 1.5 RI window and a spectra similarity score higher than 0.9 (Appendix A), and low confidence annotation requiring only a spectral match to the internal library within the RI window. Manual annotation of datasets 1 and 2 was performed by an experienced mass spectrometrist and used as a gold standard to assess the ability of WiPP to detect known compounds. For comparison, a manual annotation was generated according to our labs best practice. It consisted of data pre-processing and peak detection using Chromatof (Leco), followed by manual annotation using the in-house software, Maui-via [19]. Parameters used in Chromatof for data pre-processing are available in Appendix A.

The automated WiPP workflow achieves comparable performances to the manual annotation for medium and high concentration, as summarized in Figure 2 (full details for all compounds and concentration are shown in Appendix A). WiPP delivers 95% of the manually annotated compounds in the high concentration samples and 86% of the compounds in the medium concentration samples. However, WiPP shows some limitation with low concentration data as only 42% of the metabolites are recovered.

### 2.2. Case Study

To further validate the results produced by WiPP, we ran the full workflow, including the generation of classifiers, on a publicly available biological dataset, and results were compared to original results reported by the study. The selected study and data are introduced in Section 4.3.2. WiPP classifiers were trained using a subset of the biological samples (two samples from each biological condition) as no pooled samples were available, and peak picking algorithms parameters were optimised using a different subset of the dataset (two samples from each biological condition). The final high quality peak set obtained by WiPP was annotated using the same spectral matching similarity score, reference masses, and intensities as the original study [20]. Nine analytes were confirmed and manually validated in this study through targeted analysis. Using WiPP, six out of the nine analytes could be identified automatically with comparable calculated fold changes and corresponding *p*-values (Table 1). The remaining three analytes were identified by WiPP but labelled and flagged as shouldering peaks, requiring user attention. For these, we did not report fold changes and *p*-values because they were not automatically calculated by WiPP. Finally, one hexose, not reported in the original study, was found to be significantly different by WiPP. Our classifier labelled 100% of the peaks as high quality in all samples.

## 3. Discussion

In this study, we present WiPP, a machine learning-based pipeline that enables the optimisation, combination, and comparison of existing peak picking algorithms applied to GC-MS data. WiPP integrates machine learning classifiers to automatically evaluate the performance of peak picking algorithms and their selected parameters. WiPP also offers to the community a new approach to compare the performances of different peak picking algorithms, not only based on quantity but also on quality, and enables an automated parameter optimisation.

Our results show that WiPP produces comparable outputs to manually curate data in an automated and, thus, more reproducible and scalable manner. For low concentration peaks, the comparability is less convincing than for medium and high concentrations. This result indicates either general incapability of embedded algorithms to detect low intensity peaks or algorithm parameters to be less suitable for low intensity peaks. The latter might be the case in our setting as parameters were optimised based on samples from all three concentration ratios, therefore fitting best for the entire dataset. However, specific training and optimisation for detection of low concentration peaks would be feasible with WiPP. We consider this use case to be less relevant because low quality and low intensity peaks are also difficult to quantify and therefore add unwanted noise to statistical downstream modelling. Our main use case for WiPP currently is the automatic generation of a comprehensive set of reliably quantified high-quality peaks that is suitable for immediate downstream analysis, as needed for large scale high sample number studies. Manual inspection of intermediate quality peaks is feasible and may be useful to fully utilize a given data set, as shown in the case study. However, our case study also shows that an automated pipeline may discover features that may have escaped the attention of human specialists.

The bar plots in Appendix A show the number of peaks reported by centWave and matchFilter that are considered noise upon human inspection. At least these peaks have to be considered false positives and are of high abundance especially in the high-resolution data set. The yellow section of the bar chart in Figure 1B,D shows that substantial parts of the peaks generated by centWave and matchFilter are rejected for quality reasons. Our approach of using only high-quality peaks (methods section) aims to achieve low false positive rates. Peaks classified by WiPP with intermediate quality are more likely to be false positives, while peaks classified as noise are likely to be false positives.

### 3.1. Automated Classification of Peak Picking Provides a Novel Way to Assess and Compare the Performance of Peak Picking Algorithms

We have developed a peak quality classification system that enables algorithm-identified peaks to be classified based on user trained peak characteristics related to both peak quality and accurate quantification, e.g., apex shifted to a side, shoulder peak. We have set the number of distinct classes used for classification to 7. This is subjective but represents a balance between having enough classes to suitably differentiate between detected peaks while avoiding excessive manual annotation. Importantly, the current classifiers enable the reporting of complex peaks, such as shouldering peaks to the user for manual inspection and avoiding potential loss of data. The number of classes could be altered to suit requirements if necessary but requires some changes in the WiPP source code and will also affect the time it takes to create the training set. The time taken to manually annotate the training set is an important consideration in the functional operation of WiPP and increases proportionally with the number of classes. The manual classification of peaks still has an element of user subjectivity to it, especially where a peak may fit into more than one category (e.g., too wide and skewed). We would recommend users to be consistent in their training classification of such peaks. Future versions of WiPP may seek to address this by enabling selection of multiple peak categories for an individual peak. Model training should be carried out by someone with a good knowledge of mass spectrometry data analysis. The user will imprint his know-how and judgement into the classification model.

Our peak classification method allows us to assess the quality of peaks picked by individual algorithms, and thus enables a comparison of the relative performance of different peak picking algorithms. Peak detection in GC-MS data is a challenging and long-lasting problem. New approaches and tools emerge every year, yet there is still no established procedure to evaluate their performances objectively, and simple comparisons, such as the total number of peaks detected, is not a robust metric for benchmarking purposes [12]. It is also influenced by the selection of algorithm-specific parameters, which leads to a certain subjective component when assessing each algorithm. We have demonstrated that WiPP can objectively assess the performance of multiple peak picking algorithms and is flexible enough that new algorithms can be added by the user, thus enabling future algorithm developers to objectively rate their algorithms against competitors.

### 3.2. Optimising Parameters for Peak Picking

Currently, most peak picking algorithms require manual optimisation of parameters for every analysis. This is laborious and if not done can lead to suboptimal parameters being used to process datasets, having a strong effect on the selected peaks [18]. It is noteworthy that the heatmap figures that illustrate the parameter optimisation strategy (Appendix A) also highlight the fact that the best parameters found for matchedFilter on the considered samples do not correspond to the parameters that find the maximum number of peaks. In this specific case, the parameters displaying the highest number of peaks also find the highest number of high-quality peaks. However, it comes, at the cost of an increased number of poor quality or false positive peaks, compared to the best parameters returned by WiPP. An important consideration when dealing with poor quality peaks can be the accuracy of their integration for statistical purposes. We would argue that, in most cases where statistical analysis is being conducted on the results, it is better to have a smaller number of robust and accurately quantified peaks than a larger number of peaks with a high technical variation. Thus, we have optimised the balance between choosing the maximum number of high-quality peaks while minimising the selection of poor quality peaks. The user can decide which approach to take for themselves by changing the weighting parameters of the scoring function. Optimal parameters returned by IPO are similar to those determined by WiPP with the notable exception of the FWHM (Full Width at Half Maximum) value, which is much greater in IPO. As the average full peak width of the manually annotated peaks is 4 s, the FWHM value of 1 returned by WiPP appears to be more appropriate than the 8.8 value returned by IPO. A possible explanation is the technical differences between liquid and gas chromatography. Gas chromatography often suffers from column “bleed” at the end of an analytical run, where large amounts of chemical substances elute from the column, seen as a characteristic increase in chemical baseline noise at the end of the run. (We speculate that this well-known characteristic of gas chromatography may be distorting the ability of IPO to find an appropriate FWHM value). As IPO has been designed for liquid chromatography-mass spectrometry (LC-MS) data, it is not equipped to deal with characteristics that are specific to GC data. In our analysis, the vast majority of peaks detected after 2000 s are associated with noise and are, therefore, penalised by the WiPP optimisation approach.

### 3.3. Improving Overall Quality of the Final Picked Peak List

Interestingly, when optimised, centWave detects a higher number of true positive peaks than matchedFilter on low resolution data, while the opposite is true for high resolution data. However, it is important to note that the vast majority of peaks detected by matchedFilter on high resolution data are irrelevant (noise, duplicates, or presenting less than three characteristic *m/z*), and increases as the concentration decreases. MatchedFilter, therefore, seems better at detecting low concentration peaks, but at the expense of a higher poor-quality peak or noise selection, whereas the CentWave algorithm is better at avoiding the selection of poor quality or noise peaks, but with a potential loss of sensitivity to peaks near the signal to noise threshold. The combination of both algorithms as implemented in WiPP shows, in both low- and high- resolution data, a significant improvement on the coverage of peaks and compounds detected. These results clearly argue towards the use of several peak picking algorithms over a single one, as previously shown [21].

While only centWave and matchedFilter were integrated so far, it is possible to integrate any peak picking algorithm to the workflow to further improve the coverage of detected high quality peaks. The modular architecture of WiPP, based on the python workflow framework Snakemake, enables new peak picking algorithms integration with little effort. The more peak picking algorithms used, the longer the workflow runtime will be. Based on dataset 1 and 2 presented here, we estimate a 4 h manual peak labelling process to generate the training data per algorithm, which must only be done once. The total runtime of the workflow is highly dependent on the computing power available and the range of parameters tested. For example, the full processing for dataset 1 can be completed overnight using four cores. This time can be further reduced by narrowing the parameter search space. For high resolution data, we recommend using a high-performance computing (HPC) cluster as the number of parameters tested increases significantly.

The overall results from the benchmarking process on a known mix of commercial standards and the replication of the workflow using a publicly available dataset show that WiPP brings automated data analysis closer to the current gold standard, which is manual curation using exclusively existing peak picking algorithms. In a context where large studies become routinely run in metabolomics laboratories, it is crucial to develop automated tools that can match manually validated standards. In this respect, these results also highlight that the shortest way to automation may lie better in using existing tools than creating new ones.

We have shown that WiPP improves current automated detection of peaks by:Providing a novel way to classify peaks based on seven classes, and thus objectively assessing their quality.Enabling objective performance comparison of different peak picking algorithms.Enabling automated parameter optimisation for individual peak picking algorithms.Enabling a final, improved high quality peak list to be generated for further analyses.Reducing the operator time required by packaging WiPP within a fully automated workflow (once the initial training of data is completed).

For the definition of a peak and the deconvolution process, we needed to distinguish WiPP from the embedded peak picking algorithms. For the classification, WiPP operates on the full compound spectra, taking into account all measured m/z traces of the raw data within the peak retention time, but no deconvolution was applied at this stage. The definition of a peak and de-convolution used in the peak picking algorithms may vary for different algorithms, but WiPP is not affected by it. Indeed, this enables the combination of several peak picking approaches into WiPP. In the current implementation of WiPP, XCMS centWave and matchedFilter are used for peak detection and CAMERA [22] for the deconvolution of reported peaks. We hope that the permissive licence encourages the community to contribute by integrating additional peak picking algorithms. Compound identification is currently not a feature of WiPP and presents separate challenges. Instead, WiPP generates output files in .msp format that can be used by common library matching based identification tools. Expanding WiPP to support liquid chromatography-mass spectrometry (LC-MS) data is desirable for future versions of WiPP, as this is the technology of choice for most untargeted analysis.

## 4. Materials and Methods

### 4.1. Peak Detection

We started the WiPP workflow definition by discussing the problems of peak detection. Myers et al. have shown that, when applied to the same dataset, different peak picking algorithms return two different, yet overlapping, peak sets [8,12]. To assess the performance of a peak picking algorithm, we would like to consider the true peak set, which is unknown. The true peak set can be defined as the full set of peaks corresponding to all metabolites or contaminants (including metabolite adducts and fragments) present in a sample. The accurate quantification of a peak requires precise measurement of the peak area, which necessitates knowledge of the peak centre and boundaries. A multitude of effects, including but not limited to neighbouring peaks and technical drifts, can transform and blur the peak shape, making peak quantification problematic. This also means that there is a grey area between what is a true peak (i.e., not noise, but a distinct signal caused by a chemical) and what should be selected for further analysis. Figure 3 illustrates that algorithms select a proportion of the true peak set with a varying degree of success. For this paper, we are defining a robust/high quality peak as a peak where the peak boundaries are accurately identified and demarcated and both the signal to noise and the intensity of the peak are sufficiently high to enable accurate peak intensity measurement, allowing for robust statistical downstream analysis. Algorithms can report lesser quality measurement of true peaks (e.g., by reporting two peaks as a single peak or a single peak as two, or incorrect assessment of peak boundaries). Furthermore, each algorithm will also report “peaks” that do not correspond to the actual chemical signal (i.e., noise). Ultimately, before starting downstream analysis, user-defined filtering of what is considered “high enough” quality peaks must be defined. The schematic in Figure 3A represents chromatographic peaks being accepted or rejected by two different peak picking algorithms. Figure 3B,C illustrate that the peak sets returned by two different algorithms depend on the algorithm parameters used and have a different overlap with the actual true peak set, which, due to the addition of chemical signals from contaminants, is normally unknown, even if working with known chemical standard mixes. Maximization of the coverage of the true peak set can be achieved through an optimisation approach of the parameters of the peak detection algorithms (Figure 3C). In the same manner, the number of false positive peaks (reported by the algorithm but not corresponding to the chemical signal) returned by a peak detection algorithm also depend on the algorithm parameters. There is a trade-off between the number of false positive and false negative peaks, and users may have different preferences depending on the kind of analysis they are working on.

The quality of the picked peaks may also be parameter dependent; for example, a true peak can be reported as two separate peaks (peak splitting) or two true peaks can be reported as a single peak (peak merging). Figure 3D illustrates the objectives of WiPP, which consists of optimising parameters for multiple peak picking algorithms, classifying of the reported peaks, and combining outputs of different algorithms to automatically produce a high-quality peak set for further analysis.

#### 4.1.2. Peak Classes

For our workflow, we defined seven classes of peaks (Figure 4). Many criteria can be considered to define the different peak classes. We focused on peak shape and peak boundaries. While Figure 4 shows a schematic representation of a single *m/z* trace, WiPP operates on the full compound spectra, taking into account all measured *m/z* traces within the peak retention time. Special attention was paid to the boundaries as this heavily influences both the risk of inaccurate quantification (if peak area is used) and the risk of peak splitting and peak merging. Figure 4 shows a schematic representation of the seven classes established in WiPP, based on the selected criteria. Each of these classes carried qualitative information about the peaks. We designated classes A, B, and C as being “high quality” peaks. Class D described noise signal and was considered as a false positive, while the last 3 classes, E, F, and G represented intermediate quality true peaks, which we considered not to be robust enough for downstream analysis. In WiPP, classes E, F, and G were reported to the user for manual attention.

#### 4.1.3. WiPP Workflow and Model

The proposed workflow (Figure 5) was composed of two main distinct parts, the training of the classifiers, and the high-quality peak set generation. Supervised classifier training involved manual interaction. It should be performed at least once per instrument and sample type (i.e., blood, specific tissue, cell extract), but the same training dataset can then be used for all other analyses performed of this type. The final output of this first part was an instrument/sample type specific classifier for each individual peak detection algorithm. The second part of the workflow uses the trained classifier for unsupervised optimisation of the peak detection algorithm parameters. Subsequently, it generated a high-quality peak set based on integrating results from the individual algorithms.

The first step of the workflow aimed at generating a training peak set containing a large variety of peaks differing in quality and intensity. For this purpose, we recommend applying the algorithms to pooled or quality control samples using a wide range of parameters, and manually classifying a minimum of 700 peaks (which, in our datasets, equated to a minimum of 7 peaks in the smallest class) to generate the training set (Appendix A), henceforth called the calibration dataset. The parameter search ranges were user-defined and should be set by an experienced user with prior knowledge on both the data produced by the instrument and the algorithm they are applying (see Appendix A for the parameters used in this study). A representative set of peaks for the supervised training was generated in step 2 by sampling algorithm parameters and retention time ranges (Appendix A). WiPP provided an efficient peak visualization tool (Appendix A), allowing users to label each detected peak with one of the seven classes described in Figure 4. This important step allowed expert users to imprint their peak classification know-how into the machine learning model. The labelled peaks formed the training dataset that was used in step 3 of the workflow to train SVM classifiers. Every peak was described by an array of intensities within a certain *m/z* and retention time window. The peaks were baseline corrected, scaled, and flattened to meet the input format required by the classifiers. During training, hyper-parameter optimisation [23] was performed using stratified cross-validation to avoid over-fitting.

The fourth step of the workflow performed an unsupervised optimisation of the algorithm-specific parameters for each of the peak picking algorithms. For this purpose, the number of peaks within each class was determined and a scoring function (Appendix A) was applied that rewarded high quality peaks while penalizing low quality peaks. Relative weighting can be user-defined to cater for different use cases, e.g., discovery studies or diagnostic studies (Appendix A). To perform this unsupervised optimisation, WiPP generated a new peak set containing a large variety of peaks differing in quality and intensity. We recommend applying the peak picking algorithm using different pooled or quality control samples than the one used for the training data generation (with a minimum of two samples). We called this peak set the optimisation dataset (Figure 5). The peaks detected by every single parameter set were classified using the algorithm specific classifier and scored using the scoring function. We applied a simple grid search approach to determine the parameters returning the highest score. Those parameters were considered as optimal. We considered this method preferable to other alternatives; descent methods may lead to suboptimal solutions if the algorithm is trapped in local minima, and annealing methods are potentially computationally costly. As minima were generally shallow and broad, there was very little benefit in using more computationally costly methods. An example of the results for parameter optimisation performed by WiPP for matchedFilter based on optimising the number of peaks per quality class is shown in Appendix A.

The following step (step 5) consisted of running the peak detection algorithms with their optimal parameters on the full biological dataset.

Finally, a high-quality peak set was generated in step 6 through a series of sub-steps. First, the peaks detected by the different algorithms were classified using their respective classifiers. Next, simple filters, such as class-based removal of duplicate peaks or rejection of peaks presenting less than n *m/z,* were applied (n was set to 3 by default and could be user-defined). The resulting algorithm specific peak sets were then merged, removing duplicate peaks where the peak sets overlapped. The final peak set was composed of only high quality peaks, and the peaks predicted as low and intermediate quality were kept aside for optional further manual inspection.

### 4.2. Implementation and Availability

The pipeline was implemented in python 3 using Snakemake [24], a reproducible and scalable workflow management system. WiPP offered a modular design to allow the addition of other existing or newly developed peak picking algorithms written in common programming languages (e.g., Java, python, R). Currently, centWave and matchedFilter peak picking algorithms are available in WiPP. eRah was also considered as an additional algorithm, but, in our hands, had memory issues when processing large datasets and was thus omitted from the current release. The pipeline can be run on local computers as well as on a high-performance cluster. WiPP supports mzML, mzData, and NetCDF input formats. It was tested with Ubuntu 16 and CentOS 7.6.1810 (Core). A comprehensive user manual and quick start guide are available on the GitHub repository. WiPP is released under the permissive MIT open source license and freely available at https://github.com/bihealth/WiPP; contributions and bug reports are welcome.

### 4.3. Data

#### 4.3.1. Datasets 1 and 2

The first two datasets were made of an identical three-point dilution series (designated high, medium, and low concentration) of a compound mix of 69 metabolites in known concentrations [25]. Nine samples of each dilution (1:1, 1:10, and 1:100) for a total of 27 samples were used to form the first dataset. These samples were prepared in duplicates to be run on two instruments with different resolutions (Pegasus 4D-TOF-MS-System: Resolving Power (FWHM) = 1290 at *m/z* = 219 and 7200 Q-TOF: Resolving Power (FWHM) = 14,299 at *m/z* = 271,9867, see Appendix A for details). Sample preparation and data acquisition details are available in the Appendix A.

#### 4.3.2. Dataset 3

The third dataset was collected by Ranjbar et al. and is publicly available in the Metabolights repository [26] at https://www.ebi.ac.uk/metabolights/MTBLS105. The study evaluates changes in metabolite levels in hepatocellular carcinoma (HCC) cases vs. patients with liver cirrhosis by analysis of human blood plasma using GC-MS [20]. Briefly, data was collected using a GC-qMS (Agilent 5975C MSD coupled to an Agilent 7890A GC) equipped with an Agilent J&W DB-5MS column (30 m × 0.25 mm × 0.25 µm film, 95% dimethyl/5% diphenyl polysiloxane) with a 10 m Duragard Capillary column with a 10 min analysis, using a temperature gradient from 60 °C to 325 °C. Only 89 files generated in the selected ion monitoring (SIM) mode were used for validation purposes here. Although SIM normally simplifies peak detection, in this dataset, there were often several peaks detected for the same *m/z*, meaning that there was still a peak detection issue to be addressed.

## Figures and Tables

**Figure 1 metabolites-09-00171-f001:**
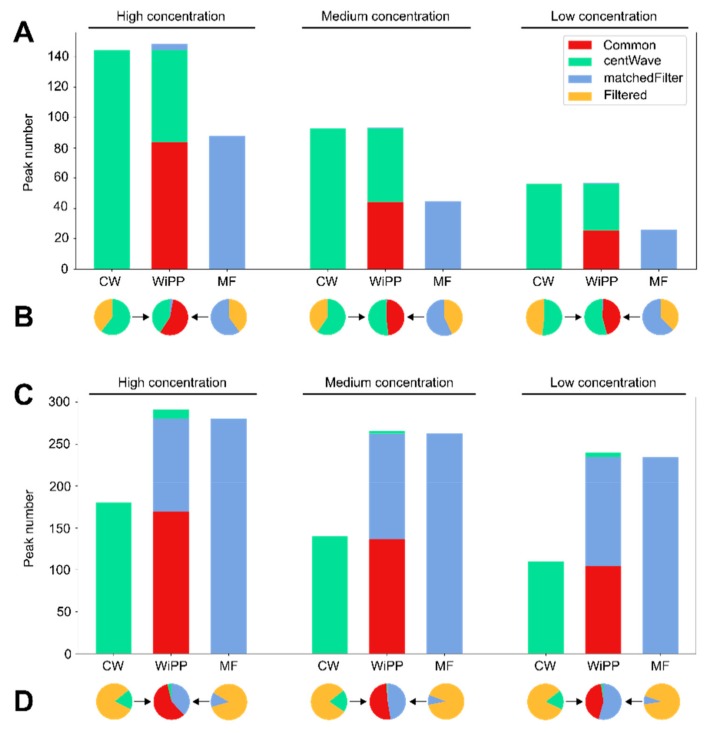
Number of peaks detected by individual algorithms on the high, medium, and low concentrations of the standard mix dataset 1 in low resolution (**A**,**B**) and high resolution (**C**,**D**). (**A**,**C**): Number of unique high-quality peaks as classified by workflow for improved peak picking (WiPP) and their algorithm of origin. (**B**,**D**): Proportion of peaks detected by centWave and matchedFilter, rejected by at least one of the quality filters in WiPP.

**Figure 2 metabolites-09-00171-f002:**
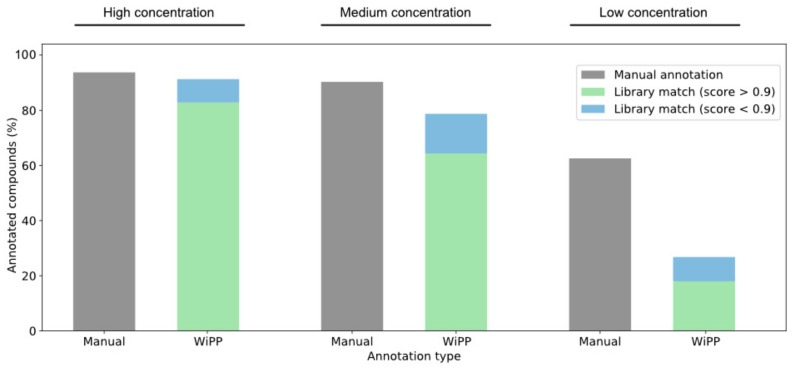
Number of metabolites detected and annotated manually or automatically by WiPP compared to the number of compounds present in the three concentrations of dataset 1.

**Figure 3 metabolites-09-00171-f003:**
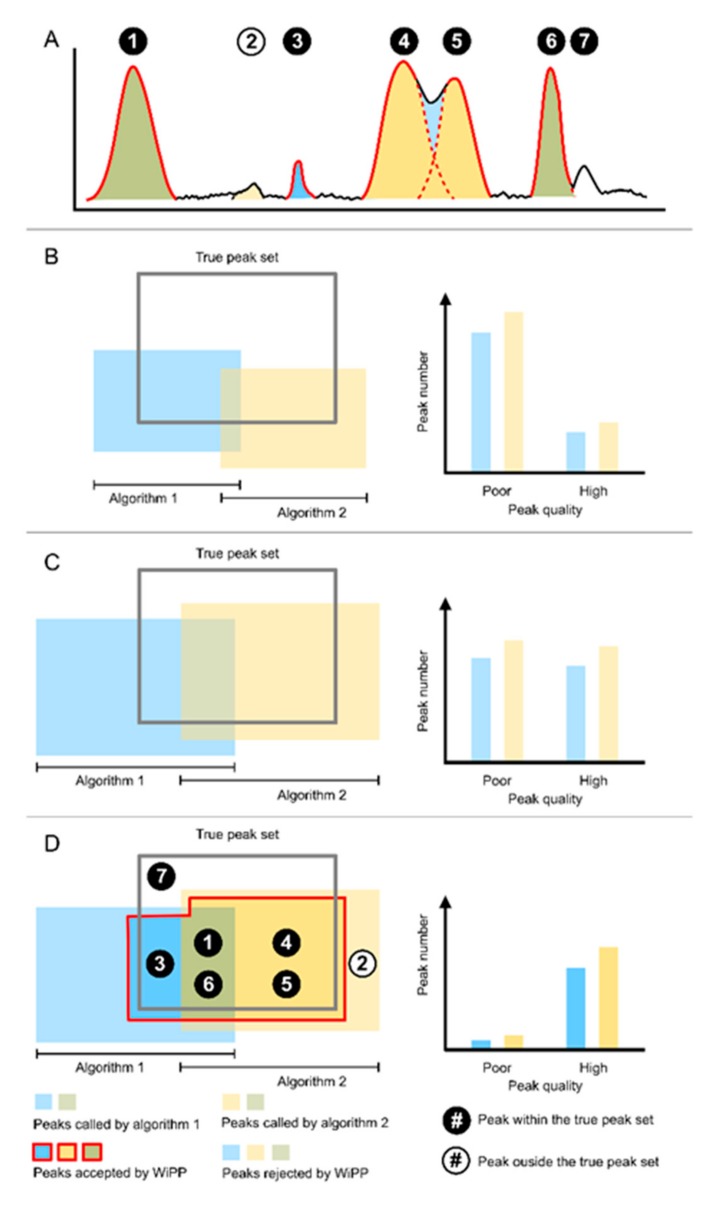
Schematic representation of the peak sets. (**A**) Chromatographic representation of peaks detected by the two peak picking algorithms accepted or rejected by WiPP. An individual ID is assigned to each peak. Peak 4 and 5 are erroneously detected by algorithm 1 as one single merged peak, hence the light blue colour between the distinct peaks properly detected by algorithm 2. (**B**) Peak called by peak picking algorithm 1 and 2 compared to the true peak set of a dataset before parameter optimisation. (**C**) Peak called by algorithm 1 and 2 after parameter optimisation. (**D**) Peaks accepted and rejected by WiPP compared to the true peak set. Circled numbers represent the peak ID from subfigure A and are placed in their respective regions in peak space.

**Figure 4 metabolites-09-00171-f004:**
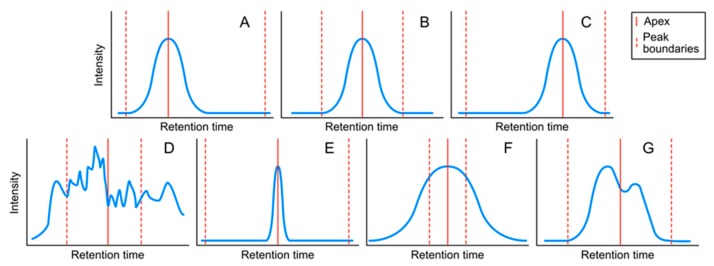
Schematic representation of the seven peak classes defined in WiPP. For clarity purposes, only one *m/z* is represented here. (**A**) Apex shifted to the left. (**B**) Centred apex. (**C**) Apex shifted to the right. (**D**) Noise. (**E**) Peak with wide margins to window borders. (**F**) Peak exceeds window borders. (**G**) Merged/shoulder peak.

**Figure 5 metabolites-09-00171-f005:**
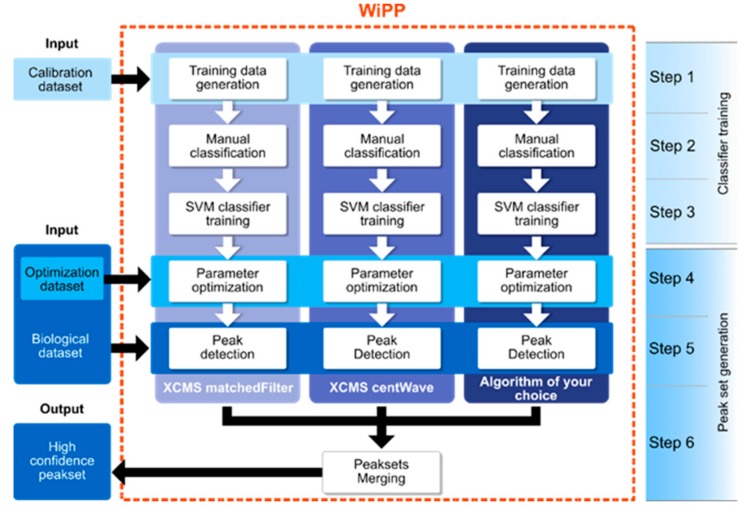
Flowchart of the WiPP method consisting of 6 steps. Steps 1 to 3 consisted of generating the training data and training the classifiers using a calibration dataset. Step 4 optimised the parameters of individual algorithms using an optimisation dataset and the trained classifiers. Step 5 ran the optimised peak detection algorithms on the full biological dataset. Step 6 classified, filtered, and merged the outputs of individual peak picking algorithms to generate a high-quality peak set.

**Table 1 metabolites-09-00171-t001:** Comparison of the results found by the published study to the one produced using WiPP automated workflow. (X) Data could not be automatically computed. (–) Missing data.

ID	Identified (WiPP)	*p*-Value (Study)	Fold Change (Study)	*p*-Value (WiPP)	Fold Change (WiPP)
Glutamic Acid	+	$5.5\times 10^{-8}$	1.9	$1.5\times 10^{-4}$	1.89
α−tocopherol	+	$1.2\times 10^{-3}$	1.5	$7.7\times 10^{-3}$	1.36
Valine	+	$3.3\times 10^{-3}$	1.5	$3.0\times 10^{-2}$	1.52
Citric Acid	+	$9.5\times 10^{-3}$	−1.3	$8.6\times 10^{-3}$	−1.20
Sorbose	+	$1.3\times 10^{-2}$	−2.4	$3.3\times 10^{-2}$	−1.66
Cholesterol	+	$3.5\times 10^{-2}$	1.1	$2.4\times 10^{-2}$	1.10
Lactic Acid	+	$2.8\times 10^{-3}$	−1.3	X	X
Leucine	+	$1.8\times 10^{-2}$	1.6	X	X
Isoleucine	+	$4.2\times 10^{-2}$	1.5	X	X
Hexose	+	–	–	$4.0\times 10^{-2}$	−1.61

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
