# Peer review of "WiPP: Workflow for Improved Peak Picking for Gas Chromatography-Mass Spectrometry (GC-MS) Data"

_metabolites, 2019, doi:10.3390/metabo9090171_

Round 1

Reviewer 1 Report

Peak detecting is one of the most important steps for GC-MS based metabolomics study. Detection of true peaks with minimum false positive peaks will largely improve data quality. In this manuscript, the authors designed a workflow called WiPP to improve the peak picking using machine learning-based classification. The topic is interesting and it provides a way to do the peak detection. There are some issues with the workflow and the presentation of the current version. The detail comments are in the following: 1. The authors argued the high false positive rates in GC-MS metabolomics peak detection require improved software. It is not clear to me whether the new software improve false positive rates or not. What are the false positive rates for the developed software and other software for the given dataset? 2. What is the metabolites identification method? With what software, or manual workflow? 3. In Fig. S4, what are the reasons for poor detection of the known metabolites in low concentration samples by WiPP? Most of the low metabolites in concentration can be detected with manual workflow. What is their signal to noise ratio? 4. In the workflow, the authors recommended to manually annotate a minimum 700 peaks to generate training dataset (line 372-374, Fig. 4 step 2). This is not acceptable for a GC-MS data processing workflow. If we need to generate such a huge reference peaks, why the users need to use this workflow? Is this step just for workflow development or for each dataset? Are the peaks detected in WiPP just within the range of the peaks manually annotated? 5. The authors used peak detection in the whole manuscript. What is the definition of a peak? For GC-EI-MS, there are so many fragments for each compound. A peak can be a compound and can be a specific m/z for a fragment. One of the most important issues for GC-MS data analysis is deconvolution or clustering the fragments into a compound. It is not clear, how does WiPP do the deconvolution step.

Author Response

Dear Reviewer 1,

Thank you very much for your valuable time and your helpful comments and suggestions. We have improved our manuscript accordingly and also addressed issues raised by the other reviewers. Please find details concerning the individual points you raised below.

To address the general improvement recommendations we have rewritten the abstract, restructured the introduction, enhanced results (new Figure 2) and methods presentation, extended the discussion and added a new section to the supplementary material. The English of the original submission had been checked by a native speaker. We try to cater for the needs of an interdisciplinary audience by rewriting many sentences. If general improvement and language concerns persist we would be grateful for examples and concrete hints.

The authors argued the high false positive rates in GC-MS metabolomics peak detection require improved software. It is not clear to me whether the new software improve false positive rates or not. What are the false positive rates for the developed software and other software for the given dataset?

The total number and rate of false positives is difficult to determine, due to the limited knowledge about the true peak set (cf. Methods). The bar plots in Figure S2 show the number of peaks reported by centWave and matchFilter that are considered noise upon human inspection. At least these peaks have to be considered false positives and are of high abundance especially in the high resolution data set. The yellow section of the bar chart in Figure 1.B and 1.D shows that substantial parts of the peaks generated by centWave and matchFilter are rejected for quality reasons. Our approach of using only high quality peaks (see methods section) aims to achieve low false positive rates. Peaks classified by WiPP with intermediate quality are more likely to be false positive, while peaks classified as noise are likely to be false positives.

We toned down the claim in the abstract and extended the discussion of this topic, lines 213-220

What is the metabolites identification method? With what software, or manual workflow?

Generally, identification/annotation of compounds is not a feature of WiPP. The compound identification methods used for the benchmarking and case study was originally described in the data section 4.3.1 and 4.3.2 and has now been moved to Results and is explained in more detail (line 148)

In Fig. S4, what are the reasons for poor detection of the known metabolites in low concentration samples by WiPP? Most of the low metabolites in concentration can be detected with manual workflow. What is their signal to noise ratio?

Thank you for the important question. We added a corresponding paragraph to the discussion, lines 202 to 212.

In the workflow, the authors recommended to manually annotate a minimum 700 peaks to generate training dataset (line 372-374, Fig. 4 step 2). This is not acceptable for a GC-MS data processing workflow. If we need to generate such a huge reference peaks, why the users need to use this workflow? Is this step just for workflow development or for each dataset? Are the peaks detected in WiPP just within the range of the peaks manually annotated?

We appreciate our description is poorly worded since it gives the impression this is a long process. Actually, the software makes manual peak quality classification very quick and easy, and the original datasets were annotated in only 3.5 hours (1 hour for 300 peaks). There is a minor gain in sensitivity between annotating 100 peaks and 700 which is shown in the supplementary figures S2. However, as this annotation is required only once per instrument and is the only step requiring manual interaction, we feel this initial investment is worthwhile. The manual classification of peaks is easy and intuitive but we recommend it is done by an expert as they will have better knowledge of what a model peak on that instrument looks like.

To reduce confusion between compound annotation and peak classification we changed the following sentence (line 401):

“[…], and manually annotate a minimum of 700 peaks to generate the training set, [...] ”

to:

“[…], and manually classify a minimum of 700 peaks to generate the training set, [...]”

The authors used peak detection in the whole manuscript. What is the definition of a peak? For GC-EI-MS, there are so many fragments for each compound. A peak can be a compound and can be a specific m/z for a fragment. One of the most important issues for GC-MS data analysis is deconvolution or clustering the fragments into a compound. It is not clear, how does WiPP do the deconvolution step. 

For the definition of a peak and the deconvolution process we need to distinguish WiPP from the actual peak picking algorithms. For the classification WiPP operates on the full compound spectra, taking into account all measured m/z traces of the raw data within the peak retention time, no deconvolution is applied at this stage. The definition of a peak and deconvolution used in the peak picking algorithms may vary for different algorithms, but WiPP is not affected by it. Indeed, this actually enable the combination of several peak picking approaches into WiPP. In the current implementation of WiPP, XCMS centWave and matchedFilter are used for peak detection and CAMERA for the deconvolution of reported peaks. To clarify this point several sentences were modified throughout the manuscript. 

Reviewer 2 Report

Manuscript: metabolites-564114

Title: WiPP: Workflow for improved Peak Picking for Gas Chromatography-Mass Spectrometry (GCMS) data
Author(s): Nico Borgsmüller , Yoann Gloaguen , Tobias Opialla , Eric Blanc , Emilie Sicard , Anne-Lise Royer , Bruno Le Bizec , Stéphanie Durand , Carole Migné , Mélanie Pétéra , Estelle Pujos-Guillot , Franck Giacomoni , Yann Guitton , Dieter Beule , Jennifer Kirwan.
The paper describes a work-flow that combines existing data processing software packages (XCMS – tat used in the experiments - metaMS, MetAlign, mzMine, ADAP-GC, PyMS and eRah) in a “smart” and interactive routine aimed at optimizing peak picking from GC-(HR)MS data set.The approach proposed includes a step where the analyst can supervise the peak detection by manually set up software specific parameters followed by an automated optimization that returns a “high” quality peak list to be further explored with chemometrics.I regret to say that the manuscript need to be re-organized and some data migrated from Supplementary to the main text. The abstract is not in line with general guidelines, it is too long and does not focuses on the core part of the work. It could be considered a summary but not an abstract.
The results section actually does not show experimental “results” and comparative performances, that are indeed reported as Supplementary material, but discusses in a very confusing and superficial way the cross-matching between tested algorithms on peak ranking.The real-analysis data set adopted to test the WiPP work-flow does not make sense to me: the training is done on a very simple mixture acquired with TOF MS at fast acquisition rate (Pegasus 4D) or in high resolution (qTOF) mode. The test set is from a GC-qMS in SIM acquisition. Why? How can be tested the results (performance data) if a third MS acquisition is introduces with a few fragments instead of a full scan spectra acquisition? This is a relevant issue that has to be re-considered before re-submission.About peak-classes I suggest authors to better define these classes using well-known chromatographic parameters as for example defining a minimal peak-width for class E, an asymmetry value for peaks A and C and a threshold or minimal SNR for peak D.
In conclusion, I think that the topic is worthy of publication, the experimental plan should be improved by adding a real-world dataset congruent with the test samples – and more realistic in a untargeted/targeted metabolomic application; the text has to be re-organized with a more logic work-flow, more didactic and understandable also to non-experts in data processing scripting but affordable for chromatographers who will be – at the end- the users of such a work-flow.Major revisions and re-submission are my indications.

Author Response

Dear Reviewer 2,

Thank you very much for your valuable time and your helpful comments and suggestions. We have improved our manuscript accordingly and also addressed issues raised by the other reviewers. Please find details concerning the individual points you raised below.

To address the general improvement recommendations we have rewritten the abstract, restructured the introduction, enhanced results (new Figure 2) and methods presentation, extended the discussion and added a new section to the supplementary material. The English of the original submission had been checked by a native speaker. We try to cater for the needs of an interdisciplinary audience by rewriting many sentences. If general improvement and language concern persists we would be grateful for examples and concrete hints.

I regret to say that the manuscript need to be reorganized and some data migrated from Supplementary to the main text. The abstract is not in line with general guidelines, it is too long and does not focuses on the core part of the work. It could be considered a summary but not an abstract.

We have appropriately rewritten the abstract

The results section actually does not show experimental “results” and comparative performances, that are indeed reported as Supplementary material, but discusses in a very confusing and superficial way the cross-matching between tested algorithms on peak ranking.

Thank you for your helpful comment. We have rewritten major parts of the results section and introduced a new figure 2 that summarizes results from the supplement.

The real-analysis data set adopted to test the WiPP workflow does not make sense to me: the training is done on a very simple mixture acquired with TOF MS at fast acquisition rate (Pegasus 4D) or in high resolution (qTOF) mode. The test set is from a GCqMS in SIM acquisition. Why? How can be tested the results (performance data) if a third MS acquisition is introduces with a few fragments instead of a full scan spectra acquisition? This is a relevant issue that has to be re-considered before resubmission.

We understand the skepticism of the reviewer which is caused by a lack of information at the adequate passage in the manuscript. The training needs to be done for each acquisition method independently as peaks depend highly on the resolution and acquisition rate. Thus, we trained for all three used data sets independent classifiers. The adaptation to a third acquisition mode further demonstrates the broad applicability of the pipeline to a variety of different GC-MS modes.

To solve this misunderstanding, we changed the description of the workflow in section 2.2

About peak-classes I suggest authors to better define these classes using well known chromatographic parameters as for example defining a minimal peak-width for class E, an asymmetry value for peaks A and C and a threshold or minimal SNR for peak D.

Classes are used to assess the shape of the peaks and quality of their extraction regardless of the parameters, if we include hard coded parameters such as minimal peak width for class E we would be replicating the parameters of the peak detection algorithm and therefore would not be able optimize them. To solve the misunderstanding we have changed the description of the class usage.

In conclusion, I think that the topic is worthy of publication, the experimental plan should be improved by adding a real world dataset congruent with the test samples – and more realistic in a untargeted/targeted metabolomic application

We appreciate the reviewer’s comments, the specific points were addressed in the previous points #3 and #4

the text has to be reorganized with a more logic workflow, more didactic and understandable also to non-experts in data processing scripting but affordable for chromatographers who will be – at the end- the users of such a workflow

We appreciate your valuable comments and have tried hard to make the paper more accessible to an broad audience. However, ultimately, it is designed as a method paper to be read by those with an understanding of the underlying bioinformatics principles. We agree that having the methods before the results would make more didactic sense, but are constrained by the limitation of the journal layout. We have enhanced the presentation substantially and included a hint for non-expert reader in the beginning of the methods section, were to read up on concepts and jargon.

An accompanying user tutorial is available on the GitHub repository which should allow also non expert users with appropriate experience in using open source bioinformatics tools to install and use the software.

Reviewer 3 Report

I was just informed that there were enough reviewers, but had already started some general comments. Here they are. A very interesting application of SVM and 

I wish the authors explained how SVM was used in more detail. It's really quite cool to think about. When I hear someone speak about SVM I picture how clouds of data are separated from each other all the way down to the actual singular values in the algorithm, and where the point or planes of separation are being drawn between clusters. In this case each point is a peak? The authors were not very clear on this more detailed understanding of SVM, although after only one read through it may be somewhere in the details I overlooked.

The other is I didn't see a full chromatogram or any mass spectra giving examples about how two algorithms differently picked a peak--using real data, not diagrams.

It is exciting as the audience for this paper seems to be pretty broad. I worry about the more biologically minded people that will not know these chemometric or practical GCMS details that would inform whomever analytical chemist they are working with to the actual performance of various instrumental and data analysis techniques when using WiPP. That way I have an idea about how to implement WiPP in a method development strategy for a laboratory.

It was a fun read, so thanks!

Author Response

Dear Reviewer 3,

Thank you very much for your valuable time and your helpful comments and suggestions. We are glad that they found our article fun.

We have improved our manuscript accordingly and also addressed issues raised by the other reviewers. Please find details concerning the individual points you raised below.

To address the general improvement recommendations we have rewritten the abstract, restructured the introduction, enhanced results (new Figure 2) and methods presentation, extended the discussion and added a new section to the supplementary material. The English of the original submission had been checked by a native speaker. We try to cater for the needs of an interdisciplinary audience by rewriting many sentences. If general improvement and language concern persists we would be grateful for examples and concrete hints.

1. I wish the authors explained how SVM was used in more detail. It's really quite cool to think about. When I hear someone speak about SVM I picture how clouds of data are separated from each other all the way down to the actual singular values in the algorithm, and where the point or planes of separation are being drawn between clusters. In this case each point is a peak? The authors were not very clear on this more detailed understanding of SVM, although after only one read through it may be somewhere in the details I overlooked.

We added a more detailed description of the SVM and our specific usage of it into the supplementary material

2. The other is I didn't see a full chromatogram or any mass spectra giving examples about how two algorithms differently picked a peak--using real data, not diagrams.

WiPP includes a visualization tool used in training, see Figure S5. We could easily include more examples from different classes but feel that this might be overwhelming for the manuscript.

3. It is exciting as the audience for this paper seems to be pretty broad. I worry about the more biologically minded people that will not know these chemometric or practical GCMS details that would inform whomever analytical chemist they are working with to the actual performance of various instrumental and data analysis techniques when using WiPP. That way I have an idea about how to implement WiPP in a method development strategy for a laboratory.

We have tried hard to deliver the topic to an interdisciplinary audience. Besides the main manuscript that tries to address all stakeholders there is technical details in the supplement. The user perspective is covered in details by the user manual and quick start manual guide available in the github repository.

Round 2

Reviewer 1 Report

The revised manuscript is much better than the original version. The authors addressed the issues the reviewers pointed out. This paper focuses on the peak picking. However, there are other important steps to make a full flow-chart for metabolomics study such as automation of reference compound list, metabolites identification, and retention index calculation.  It would be great that the authors make a paragraph to describe the limitation of the current pipeline and the future work.   

Author Response

Dear Reviewer,

We agree that there are many other important steps to be covered in a metabolomics data analysis and that every step presents its own challenges that must be addressed. The current version of WiPP aims at improving the important peak-picking step. However, we intend to further expand its scope and performance according to user feedback. We have added a paragraph on limitation and possible direction of future work at the end of the discussion section.   

Thank you again for your much-appreciated comments and hints, that enabled us to improve the manuscript.

Reviewer 2 Report

The present form can be accepted for pubblication. The reorganization had impacted on the readability and understanding.

Author Response

Thank you again for your much-appreciated comments and hints, that enabled us to improve the manuscript.